# Projection-based stereolithography for direct 3D printing of heterogeneous ultrasound phantoms

Samantha J. Paulsen[1☯], Trevor M. Mitcham[2,3☯], Charlene S. Pan[1], James Long[2], Bagrat Grigoryan[1], Daniel W. Sazer[1], Collin J. Harlan[2,3], Kevin D. Janson[1], Mark D. Pagel[3,4], Jordan S. Miller[1]*, Richard R. Bouchard[2,3]*

1 Department of Bioengineering, Rice University, Houston, TX, United States of America, 2 Department of Imaging Physics, Division of Diagnostic Imaging, The University of Texas MD Anderson Cancer Center, Houston, TX, United States of America, 3 The University of Texas MD Anderson Cancer Center Graduate School of Biomedical Sciences, Houston, TX, United States of America, 4 Department of Cancer Systems Imaging, Division of Diagnostic Imaging, The University of Texas MD Anderson Cancer Center, Houston, TX, United States of America

☯ These authors contributed equally to this work.
* rrbouchard@mdanderson.org (RRB); jmil@rice.edu (JSM)

**Data Availability Statement:** Data are available from Figshare (https://figshare.com/projects/ Projection-based_stereolithography_for_direct_

## Abstract

Modern ultrasound (US) imaging is increasing its clinical impact, particularly with the introduction of US-based quantitative imaging biomarkers. Continued development and validation of such novel imaging approaches requires imaging phantoms that recapitulate the underlying anatomy and pathology of interest. However, current US phantom designs are generally too simplistic to emulate the structure and variability of the human body. Therefore, there is a need to create a platform that is capable of generating well-characterized phantoms that can mimic the basic anatomical, functional, and mechanical properties of native tissues and pathologies. Using a 3D-printing technique based on stereolithography, we fabricated US phantoms using soft materials in a single fabrication session, without the need for material casting or back-filling. With this technique, we induced variable levels of stable US backscatter in our printed materials in anatomically relevant 3D patterns. Additionally, we controlled phantom stiffness from 7 to >120 kPa at the voxel level to generate isotropic and anisotropic phantoms for elasticity imaging. Lastly, we demonstrated the fabrication of channels with diameters as small as 60 micrometers and with complex geometry (e.g., tortuosity) capable of supporting blood-mimicking fluid flow. Collectively, these results show that projection-based stereolithography allows for customizable fabrication of complex US phantoms.

## Introduction

Ultrasound (US) imaging has long been a valuable tool for medical diagnostics due to its non-invasive nature, high resolution, dynamic-imaging capabilities, and its capacity to assess tissue properties beyond simple anatomy (e.g., blood flow or tissue stiffness) [1]. In addition to

3D_printing_of_heterogeneous_ultrasound_
phantoms/126268).

**Funding:** This work was supported in part by the U.
S. National Heart, Lung, and Blood Institute of the
National Institutes of Health F31 NRSA Fellowship
(S.J.P., HL134295; https://www.nhlbi.nih.gov/
grants-and-training/training-and-career-
development); the Rice University Special Endowed
Nettie S. Autrey Fellowship (S.J.P.; https://
graduate.rice.edu/financialsupport); the National
Science Foundation Graduate Research Fellowship
(B.G., 1450681; https://www.nsfgrfp.org/); and a
training fellowship from the Gulf Coast Consortia
on the NSF IGERT: Neuroengineering from Cells to
Systems (D.W.S.,1250104; http://www.
gulfcoastconsortia.org/home/training/
neuroengineering-igert/); the National Cancer
Institute of the National Institute of Health R21
Exploratory/Developmental Research Grant Award
(RRB; 1R21CA234526; https://grants.nih.gov/
grants/funding/r21.htm). The funders had no role
in study design, data collection and analysis,
decision to publish, or preparation of the
manuscript.

**Competing interests:** The authors have declared
that no competing interests exist.

traditional anatomy-based US applications, the use of US-based quantitative imaging bio-markers (QIB)–such as assessing volume blood flow (VBF), tissue perfusion with contrast-enhanced US (CEUS), or shear wave speed (SWS) elasticity imaging–offers tremendous potential in providing more effective, patient-specific, rational clinical care [2, 3]. However, translating QIB methods from research tools to clinical practice has proven challenging, in large part because the imaging phantoms needed to support robust quality assurance (QA) programs for these modalities are often insufficient. Given the strong dependence of QIBs on the specific functional (e.g., blood flow) and anatomical (e.g., vessel topology) aspects of an interrogated biological system, current imaging phantoms often do not adequately simulate the wide range of structural complexities and biological variation inherent in a human subject due to their overly "simplistic design." This leads to overestimated measurements for precision, particularly reproducibility, and inaccurate assessment of bias [4–7]. Studies have noted that many physiological (e.g., vessel permeability) and anatomical (e.g., vessel scale and dimensionality) characteristics are not accurately replicated in current imaging phantoms [8, 9]. The RSNA QIB Alliance (QIBA) Metrology Working Group recently warned that "phantoms do not represent the complexity of human targets; thus, precision is often overestimated" [10]. Consequently, they claimed that "improved realism of phantoms [is an] area worthy of further investment" [10]. To this end, an improved phantom platform is critical for validation and optimization of more established US-based QIB methods (e.g., VBF, CEUS, & SWS), while such a platform would also be of tremendous value in the development of newer QIB approaches, such as super-resolution imaging, acoustic angiography, or photoacoustic-US oxygen saturation imaging. To meet these needs, the next-generation imaging phantom platform should be capable of capturing the scale, tortuosity, density, and functionality of vasculature and of emulating the tissue backscatter heterogeneity, viscoelasticity, and anisotropy that is characteristic of human biology.

Although homogeneous tissue-mimicking phantoms made of hydrogels, rubbers, and other tissue-mimicking materials are widely available, both commercially (e.g., CIRS, Nuclemed, True Phantom) and otherwise, most provide only gross, organ-level anatomy- and physiology-mimicking properties [11–15]. While such tissue-mimicking extent is generally sufficient for routine imaging and system QA testing, these phantoms ultimately lack the heterogeneity and functional aspects (e.g., realistic blood flow) ideal for use with US-based QIB imaging. For instance, phantoms for Doppler imaging and elasticity imaging should include flow-supporting channels and regions of varying stiffness, respectively. Unfortunately, there currently exist limitations in fabricating realistic and well-characterized phantoms with the degree of spatial control necessary to make such voxel-specific changes in desired phantom properties.

To address the traditionally limited geometric complexity in phantoms, researchers have investigated tissue-mimicking phantoms via additions of materials such as tube-like structures and inclusions of varying backscatter and/or stiffness to homogeneous tissue-mimicking phantom bases. However, such processes are generally time and labor intensive (i.e., require multiple casting sessions), are generally compatible with only basic geometries, and often introduce imperfections (e.g., air, other unintended materials) that generate imaging artifacts [16–20]. Similar methodology has been used to create tissue-containing phantoms that are anatomically realistic at both macro- and micro-scales but whose use is generally limited owing to their relative lack of characterization compared with wholly fabricated phantoms [21, 22].

Three-dimensional (3D) printing can address many of these fabrication challenges by giving researchers control over every voxel within the print volume [22]. Indeed, multiple groups have used 3D printing to develop custom molds for generating US phantoms that mimic

patient anatomy. As an example, fused deposition modeling (FDM) has been used to fabricate plastic molds for casting US phantoms of the thyroid [23], a fluid-flow phantom [24], and phantoms with bone-like inclusions to mimic the spine [25] and rib cage [26]. This process can also be used to create molds of individual tissue-mimicking components with varying levels of backscatter, which are then combined to produce US phantoms mimicking whole organs, such as the human heart or placenta [27]; however, these processes inherently require significant time and effort while limiting the ultimate phantom complexity (i.e., variation in sub-voxel structural, acoustic, and stiffness properties) that is reasonably achievable.

The recent expansion of 3D printing techniques beyond plastic-based FDM has enabled researchers to fabricate phantoms directly from soft materials. For example, commercially available inkjet-based printing systems have been used to directly fabricate silicone-based models of abdominal aortas for inclusion in US phantoms. Although promising, these phantoms were orders of magnitude stiffer than typical soft tissue, with storage moduli on the order of 1 MPa [28]. Other groups have used inkjet technology to generate regions of binary hyperechogenicity at the imaging voxel level; however, the need for a support material during the fabrication process ultimately limits control over the phantoms' contrast and spatial resolution [29, 30]. Additionally, several tested materials do not provide adequate US image quality to qualify as potential tissue-mimicking phantoms [31, 32]. Therefore, challenges remain with respect to 3D printing phantoms using soft materials with varying levels of backscatter or stiffness in a single fabrication session.

To improve the quality and ease of fabrication of phantoms for a wide range of US-mediated imaging modalities, we used projection-based stereolithography (pSLA) to fabricate phantoms containing voxel-specific US backscatter, direct-printed targets for elasticity imaging, and with open-channel networks for high-resolution Doppler imaging. We previously used a pSLA technique developed to print binary structures using poly(ethylene glycol) (PEG)-based hydrogels [33]. While this class of hydrogel has been shown to generate material with US properties (i.e., speed of sound, attenuation) generally in the range of human tissues [34], this previous work did not explore formulations and fabrications designed exclusively for the purpose of making the specimen amenable for high-quality, clinical US imaging. In this work, we incorporated an additional functionality into our existing 3D-printing technique to vary the levels of light exposure within each printed layer and voxel, thereby permitting fine spatial control of US backscatter and stiffness. Our semi-automated approach can print at rates of up to 3 cm (in the Z axis) per hour, permitting the timely production of fully customized and cured imaging phantoms. Ultimately, our method provides researchers with the ability to develop phantoms with customized backscatter and elasticity values as well as with complex vasculature such that they may be made sufficiently realistic and detailed to generally mimic tissue for further development and validation of functional and QIB US imaging techniques.

## Materials and methods

### Hydrogel materials and synthesis

PEG diacrylate (PEGDA) of molecular weight (MW) of 6 or 35 kDa was synthesized, as described previously [35]. Briefly, PEG of the desired MW was reacted with triethylamine and acryloyl chloride in dichloromethane under anhydrous conditions with argon overnight. Lithium phenyl-2,4,6-trimethylbenzoylphosphinate (LAP) was prepared, as described previously [36]. Briefly, dimethyl phenylphosphinite was reacted with 2,3,6-trimethylbenzoyl chloride under argon overnight at room temperature. A 4-molar excess lithium bromide in 2-butanone was added to the mixture, which was then heated to 50°C to allow the formation of a solid

precipitate. The mixture was cooled to room temperature for 4 hours and then filtered with excess 2-butanone and diethyl ether. For phantom samples used in the stability experiment, a gelatin methacrylate (GelMA) additive was synthesized, as described previously [33]. PEGDA and GelMA macromers were dissolved in phosphate-buffered saline (PBS) to make pre-polymer solutions. Using [1]H NMR [35], percent acrylation of PEGDA was determined to be 99%, and yields generally ranged from 80–90% for batch sizes of up to 350 g. For LAP preparation, yields of up to 90% were achieved for batch sizes of up to 30 g.

## Design and fabrication of phantoms

US phantoms were fabricated using a pSLA system, as described previously [33]. Briefly, this 3D printing system consisted of a Z-stage print platform opposing a transparent PEGDA resin basin coated with Polydimethylsiloxane (PDMS) to avoid PEGDA adhesion and a PRO4500 optical engine emitting a 405-nm ultraviolet light (Wintech Digital Systems Technology Corp., Carlsbad, CA), achieving a nominal 50-μm print resolution, and allowing a maximal print volume (X/Y/Z) of 50x40x65 mm. Note that Phantom 15, which contained small channels capable of supporting flow, was fabricated with a higher-resolution pSLA system that used a PRO6500 optical engine (Wintech Digital Systems Technology Corp.) that can achieve a nominal 10-μm print resolution. Phantoms were fabricated layer by layer (Fig 1) from a liquid pre-polymer solution that solidified upon exposure to light. During the 3D printing process, light absorption and scattering through the layers caused an exposure gradient, and therefore a gradient in PEGDA photocuring, which manifested as a discontinuity of local crosslinking density at layer boundaries (Fig 1C) [37]. This crosslinking discontinuity consequently results in a mass

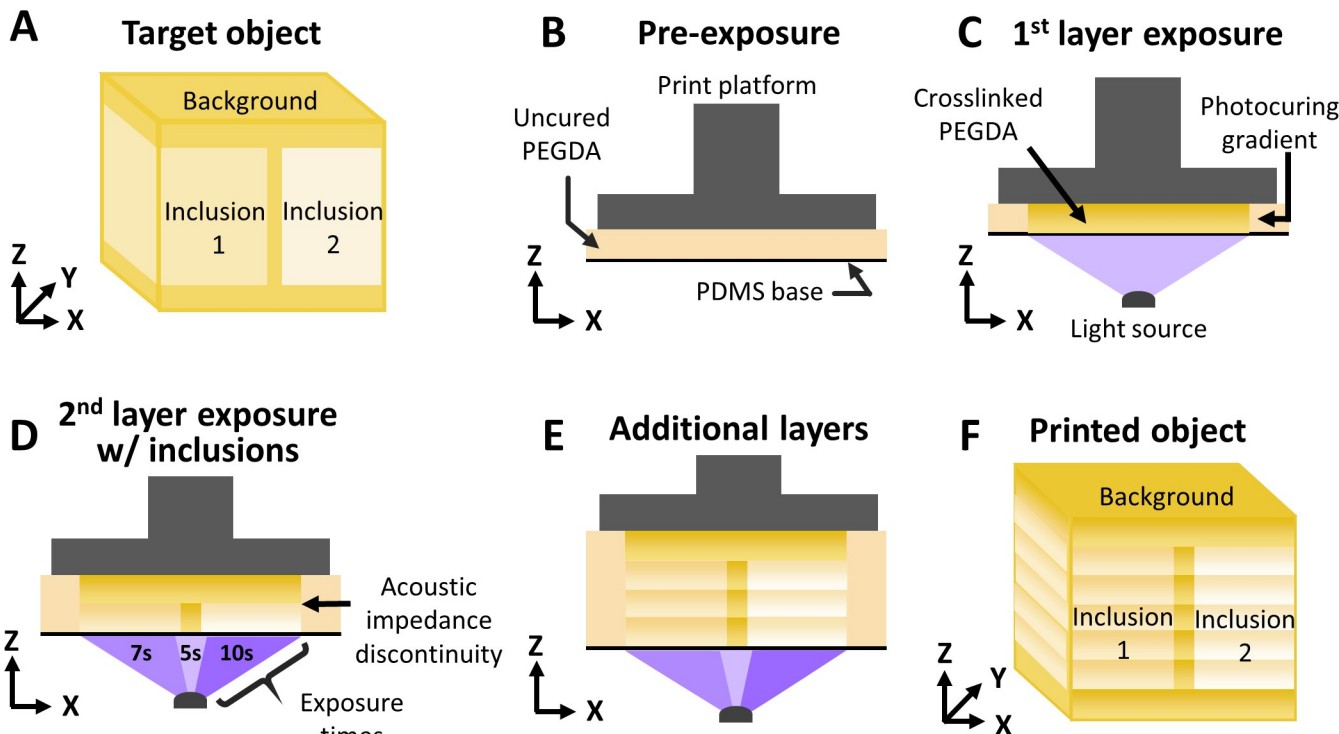

**Fig 1. 3D printing of phantoms.** A CAD model was translated into (**A**) the desired phantom. (**B**) PEGDA material was extruded in layers of set thicknesses and (**C**) photocured with ultraviolet light (purple) from the bottom. (**D**) Inclusions were incorporated within the PEGDA material by spatially varying light exposure time. (**E**) Subsequent layers were then extruded and photocured using the same technique until (**F**) the physical phantom was fabricated.

density discontinuity (i.e., acoustic impedance mismatch), generating varied levels of US back-scatter. Additionally, as elasticity and crosslinking density are directly linked, this printing method provides a mechanism for creation of tunable elasticity phantoms based on differing photocuring times.

The desired geometries of the US phantoms were designed using SolidWorks (Dassault Systems, Velizy-Villacoublay, France) to generate computer-aided design (CAD) models for each test sample. The printer control software, Creation Workshop (EnvisionTEC, Inc., Dearborn, MI), was used to slice 3D models into layers and generate the machine code (G-code) to control the position of the Z-platform. All hydrogels were fabricated using a pre-polymer solution containing varying concentrations of PEGDA as a base, tartrazine as a photoabsorber, and with 34 mM LAP as a photoinitiator (17 mM LAP for the GelMA formulation). The concentration of PEGDA was modified to impart different mechanical properties to the resulting hydrogels, and the concentration of tartrazine was modified depending on the thickness of the print layers. Inclusion of GelMA was investigated given its demonstrated impact on the swelling and stability behavior of constructs when mixed with PEGDA [38]. Finally, the per-layer exposure time was also modified depending on the PEGDA concentration and layer thickness. Printing parameters for each phantom are outlined in Table 1. Gels were stored in either deionized (DI) water or PBS prior to imaging. This rehydration process also flushed any

**Table 1. Description of hydrogel formulations, additives and print parameters for phantoms.**

| | Phantom Description | Figure | Layer (µm) | PEGDA Formulation | Additives | Tartrazine (mM) | Cure Time (sec) | Print Time (min) |
|---|---|---|---|---|---|---|---|---|
| 1 | 200-µm Layer | 2, 3, S2 | 200 | 20 wt% 6 kDa | None | 1.4 | 4 | 5 |
| 2 | 50-µm Layer | 2, 3, S2 | 50 | 20 wt% 6 kDa | None | 2.15 | 8.5 | 35 |
| 3 | 200-µm Optical | 2, 3 | 200 | 20 wt% 6 kDa | 1.0 mg/mL FITC-Dextran (150 kDa) | 1.4 | 4 | 5 |
| 4 | 50-µm Optical | 2, 3 | 50 | 20 wt% 6 kDa | 1.0 mg/mL FITC-Dextran (150 kDa) | 2.15 | 8.5 | 35 |
| 5 | Additive-free Contrast | 2, 3, 4 | 50 | 20 wt% 6 kDa | None | 2.15 | 9.5 (Base)[1] | 24 |
| 6 | Xanthan Contrast | 2, S5 | 50 | 20 wt% 6 kDa | 0.833 mg/mL Xanthan Gum | 2.15 | 9.5 (Base)[1] | 24 |
| 7 | Xanthan and Low Silica Contrast | 2, 4, S5 | 50 | 20 wt% 6 kDa | 0.1 mg/mL Silica & 0.833 mg/mL Xanthan Gum | 2.15 | 9.5 (Base)[1] | 24 |
| 8 | Xanthan and High Silica | S5 | 50 | 20 wt% 6 kDa | 1.0 mg/mL Silica & 0.833 mg/mL Xanthan Gum | 2.15 | 9.5 | 24 |
| 9 | Speed of Sound | 2 | 100 | 20 wt% 6 kDa | None | 2.25 | 11 | 35 |
| 10 | PEGDA Stability | 2 | 100 | 20 wt% 6 kDa | None | 2.25 | 11 | 11 |
| 11 | PEGDA-GelMA Stability | 2 | 50 | 3.25% 3.4 kDa | 10% GelMA & 10% glycerol | 2.25 | 8 | 17 |
| 12 | Compliant Elasticity | 2 | 50 | 20 wt% 1:3 6:35 kDa | None | 2.15 | 5 (Base)[2] | 36 |
| 13 | Stiff Elasticity | 2 | 50 | 20 wt% 6 kDa | None | 2.15 | 8 (Base)[3] | 54 |
| 14 | Anisotropic Elasticity | 2, S1, S3 | 50 | 20 wt% 1:3 6:35 kDa | None | 2.15 | 5 (Base) 7.5 (Stripes) | 28 |
| 15 | Small-Channel Flow | 5 | 50 | 20 wt% 6 kDa | None | 2.81 | 12.5 | 15 |
| 16 | Serpentine Flow | 5, S4 | 50 | 80 wt% 1:1 6:35 kDa | None | 2.15 | 3.5 | 6 |
| 17 | Tumor Flow | 6 | 50 | 80 wt% 1:1 6:35 kDa | 0.1 mg/mL Silica 0.833 mg/mL Xanthan Gum | 2.15 | 6 (Base) 2 (Tumor) | 25 |

1. Additional targets were printed with 7 cure times (10, 10.5, 12, 14.5, 19.5, 24.5, & 39.5 total seconds) within the phantom.

2. Additional targets were printed with 3 cure times (7.5, 10, & 12.5 total seconds).

3. Additional targets were printed with 3 cure times (12, 17, & 20 total seconds).

unreacted solution from vessel lumen and washed the phantom of tartrazine to provide transparent media.

Local US backscatter was generated through voxel-level (acoustic) impedance mismatches at print-layer boundaries. It is important to note that most of the printed backscatter results from specular scattering, as adjacent print voxels were generally photocured to present with the same impedance mismatch. But because adjacent 50-μm print voxels (i.e., below the typical resolution for clinical US) can be made to have varied mismatches (e.g., the anisotropic elasticity phantom), it is possible to generate diffuse scattering. However, the *print* voxels' relative size and regular periodicity severely limit their ability to generate fully developed speckle, which requires highly dense acoustic scatterers of random phase/amplitude within an *imaging* voxel. Therefore, we fabricated multiple phantoms (Phantoms 7 and 17) with 0.1 mg/mL of 40-μm diameter silica particles (MIN-U-SIL-40, U.S. Silica Co., Mill Creek, OK) and one phantom (Phantom 8) with 1.0 mg/mL of silica particles to provide US speckle more typical of soft tissue, as has been demonstrated in previous work of conventional gelatin-based phantoms [12].

## US imaging setup

All imaging studies–except for those involving elasticity imaging–were conducted on a Vevo 2100 US system (FUJIFILM VisualSonics Inc., Toronto, Canada) using MS200 (9–18 MHz bandwidth), MS400 (18–38 MHz), or MS700 (30–70 MHz) linear array transducers. This system has a built-in stepper motor to translate the transducer in the elevation dimension to acquire 3D imaging data. Beamformed imaging data were then exported into MATLAB (Mathworks, Natick, MA) for analysis and visualization. US elasticity imaging was performed on a Vantage 128 system (Verasonics, Inc., Kirkland, WA) using an L11-4v (4.5–11 MHz) linear array transducer. For all imaging studies, the printed phantom was secured to a gelatin (8 wt%) base using cyanoacrylate glue to mitigate reverberation artifacts at the distal edge. The phantom was then coupled to the transducer via a PBS or deionized (DI) water bath, except for Doppler imaging studies, in which US gel was used for coupling.

## Assessment of US properties and stability

To investigate the US properties of the printed phantoms, we acquired volumetric B-mode US data of phantoms with different print-layer thicknesses (Phantoms 1,2; Fig 2A) using 12

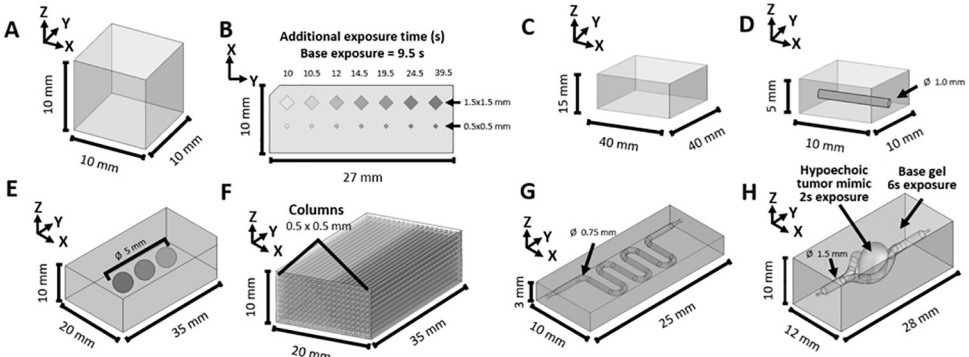

**Fig 2. Imaging phantom schematics.** (**A**) Layer thickness phantom (Phantoms 1–4). (**B**) Contrast phantoms with printed inclusions of different sizes and photocuring times (Phantoms 5–7). (**C**) Speed of sound phantom (Phantom 9). (**D**) Temporal stability phantom (Phantoms 10,11). (**E**) Elasticity phantom with three inclusions of increased photocuring time (Phantoms 12,13). (**F**) Anisotropic elasticity phantom with columns printed in a checkerboard pattern in the X-Z plane (Phantom 14). (**G**) Serpentine flow phantom (Phantom 16). (**H**) Tumor flow phantom with a vascular-mimicking branched channel and a tumor-mimicking inclusion in the center (Phantom 17). Note that ∅ denotes object diameter.

(MS200), 30 (MS400), & 50 (MS700) MHz center frequencies. In addition, optically translucent phantoms were fabricated with 1 mg/mL of 150 kDa fluorescein isothiocyanate (FITC) −dextran (MilliporeSigma, Burlington, MA), an agent known to photobleach rapidly, added to the print solution (Phantoms 3,4). These phantoms were imaged with fluorescence microscopy using a TE Eclipse epifluorescence microscope (Nikon Instruments Inc., Melville, NY) for comparison with acquired US images.

To further characterize US signals generated within the phantoms, B-mode volumetric imaging was performed with the aforementioned transducers/frequencies to assess all 7 levels of varied photocuring time within the contrast phantoms (Phantoms 5–7; Fig 2B), which consisted of rectangular patterns within the X-Y plane. A custom Python script was developed to alter the order of G-code so that certain shapes of selected layers received a secondary light exposure before proceeding to the next Z-position (S1 Fig). Xanthan gum, a common food additive, was added in some phantoms to prevent silica particles from settling during the printing process. B-mode images (30 MHz; MS400) of Phantoms 6–8 were qualitatively assessed to visualize the difference in scatterer distribution across multiple silica concentrations. In addition, because PEGDA degrades via hydrolysis, we assessed the temporal stability of the US signal from the printed targets (i.e., regions with additional cure time) and the base hydrogel (i.e., the region with no additional cure time) over 6 weeks. Matched acquisitions were performed 1, 3, & 6 weeks after baseline imaging following fabrication to assess imaging temporal stability. All imaging parameters (e.g., gain, transmit power, TGC, dynamic range, and position within the field of view) remained constant across imaging sessions. Signal and contrast-to-noise ratio (CNR) were analyzed from the B-Mode US data across five slices in each printed target for all 4 time-points via the establishment of regions of interest (ROIs) within each printed target and matched ROIs in the base hydrogel region. Signal was reported as the mean within the ROI, whereas CNR was defined as [39].

$$CNR = \frac{\mu_{signal} - \mu_{background}}{\sqrt{\sigma_{signal}^2 + \sigma_{background}^2}}, \tag{1}$$

where $\mu_{signal}$ is the mean B-mode signal within the printed-target ROI, $\mu_{background}$ is the mean signal within an ROI outside of the printed target, $\sigma_{signal}$ is the standard deviation of the signal within the printed-target ROI, and $\sigma_{background}$ is the standard deviation within the ROI outside the printed target. Using this method, we compared matched ROIs in the printed target and a region of equal area in the phantom background at equal depth.

To assess print stability and fidelity, rectangular phantoms with a 1mm-inner-diameter linear lumen (Phantoms 10,11; Fig 2D) were printed and immediately stored in DI water or PBS at 4 or 25°C. In addition to a standard PEGDA formulation (Phantom 10), a second PEGDA formulation containing GelMA (Phantom 11) was also tested. B-mode volumetric imaging (30 MHz; MS400) was performed on matched pairs (i.e., duplicate phantoms) of each formulation stored at each temperature, with the phantom lumen positioned perpendicular to the transducer axis for four time-points: 0 (i.e., immediately after printing), 1, 2, & 4 weeks following printing. For each time-point, three imaging slices separated by 3–4 mm were chosen, and the midsection extent in the Y- and Z-axis of the rectangular phantom's walls and lumen was measured using digital calipers in the VevoLAB analysis package (FUJIFILM VisualSonics, Toronto, ON, Canada). Precision was assessed by the standard deviation of measurements for a constant geometric feature (e.g., lumen) across three locations for the same phantom; reproducibility was assessed by comparing the percent difference in geometric features between two matched phantoms; stability was assessed by tracking the change in geometric features for a

given phantom over time; and accuracy (i.e., print fidelity) was assessed by comparing the dimensions of printed GelMA-containing phantoms to their CAD dimensions.

To assess the speed of sound through phantom samples at US frequencies, a uniform rectangular phantom with no inclusions (Phantom 9; Fig 2C) was placed on a stainless-steel plate and secured with a Saran plastic wrap (S. C. Johnson & Son, Inc., Racine, WI) strip (i.e., the width of the phantom) overtop. The plate was then put in an UMS Research US measurement water tank containing a 2-axis translation stage with transducer holder (Precision Acoustics Ltd., Dorchester, England); the tank was filled with degassed DI water at a temperature of 20.0 ˚C. Using a UT320 pulser-receiver (UTEX Scientific Instruments Inc., Mississauga, Ontario, Canada) connected to a DSOX3024A oscilloscope (Keysight Technologies, Santa Rosa, CA), an US pulse was transmitted and received via an unfocused, circle transducer positioned 25 mm above a region of the plate not containing the phantom. The transducer was then translated laterally to five regions (separated by ~3 mm) overhead the phantom sample, and the pulse-echo scheme was repeated at each. Speed of sound through the sample was then determined using the pulse-echo substitution method [11]. The following unfocused, circle immersion transducers (UTX, Inc., Holmes, NY) were used with these center frequencies (MHz), element diameters (in.), & -6dB bandwidth (%): 2.25, 0.5, & 85; 5.0, 0.5, & 73; 10.0, 0.375, & 91.

## Elasticity imaging

To create elasticity imaging phantoms, two different PEGDA formulations with printed 5-mm spherical inclusions (Fig 2E) were used: one (Phantom 12) with a base cure time of 5 seconds containing three printed spherical targets with total cure times of 7.5, 10, & 12.5 seconds and one (Phantom 13) with a base cure time of 8 seconds with 3 targets exposed for 12, 16, or 20 seconds. Imaging and analysis code was adapted from Deng *et al*. for acquisition [40]. Briefly, an acoustic radiation force impulse excitation "push" of 900 cycles at a 6.25-MHz center frequency was focused at a 20-mm depth within the phantom, 5-mm laterally from each target (Fig 2E), to generate a shear wave that was then tracked with pulse-echo US to determine local SWS of the material. Shear wave velocity through the phantom background and the inclusions was measured [39, 41] to obtain Young's modulus estimates assuming a linear, isotropic, elastic medium. Each phantom was imaged 10 times for statistical analysis.

We fabricated a third elasticity imaging phantom (Phantom 14) to demonstrate transverse anisotropy of shear wave propagation. While the background of the phantom was cured for 5 seconds, columns running throughout the phantom in a checkerboard fashion were cured for a total time of 7.5 seconds (Fig 2F). The phantom was then placed on a rotating base to allow for image acquisitions at 0˚, 45˚, & 90˚ angular positions. Shear wave imaging was performed in the same manner as described previously, beginning with the transducer's long axis oriented parallel to the axis of the printed column targets (i.e., 0˚; Fig 2F). The phantom was then rotated to create a 45˚ angle between the column targets and the transducer and then a 90˚ angle (i.e., transducer's long axis perpendicular to the column targets), and shear wave imaging was performed 10 times at each angular position.

## 3D flow imaging

Doppler imaging was used to assess the printing technique's ability to generate flow-supporting channels (Phantom 15). To this end, we infused 3-μm polystyrene beads (Magsphere Inc., Pasadena, CA) diluted at a ratio of 1:15 in 25% glycerol solution as flow imaging targets [42]. This suspension was then infused into the phantom using a syringe pump (New Era Pump Systems, Inc., Farmingdale, NY) at flow rates of 100, 25, 25, & 10 μL/min for CAD-based lumen

diameters of 500, 300, 250, & 150 μm, respectively. Note that tartrazine concentration was increased to 2.81 mM for this phantom to increase light absorption and thus minimize unwanted curing from the stronger backlight of the high-resolution projector, while layers were printed perpendicular to the channel axis to limit light penetration into the channels. The phantom was coupled to the transducer with US gel to prevent water coupling into the channels, and color Doppler US imaging (30 MHz; MS400) of the flowing beads was performed using a pulse repetition frequency (PRF) of 2–3 kHz. Upon validation of flow throughout all phantom channels, channel diameters were measured from B-mode images with the VevoLAB analysis package's digital calipers; the diameter at the narrowest measured point within the channel was recorded.

Next, a phantom was fabricated to image flow through a planar 'serpentine' channel architecture containing a single channel, with a nominal 1-mm diameter with straight sections separated by 180˚ turns (Phantom 16; Fig 2G). It was perfused with the same glycerol solution at a flow rate of 100 μL/min, and then color Doppler imaging data (1-kHz PRF; 24 MHz; MS400) were acquired for the purpose of reconstructing the 3D-flow vector components. The phantom was placed on the rotating base to provide 0˚, 45˚, & 90˚ acquisition angles with both +15 and -15˚ beam steering angles at each rotation angle (i.e., six unique views for each voxel). After imaging, Doppler values from all six acquisitions were imported into MATLAB and spatially co-registered to six degrees of freedom with an affine transform to obtain the X-, Y-, & Z-vector components for flow velocity at each voxel.

Finally, we fabricated an anatomy-mimicking vascularized tumor phantom with a hypoechoic tumor-mimicking printed target flanked by a 3D branched vessel network (Phantom 17) that included a common entry channel that branched into two distinct 3D vessels (Fig 2H). Note that Phantoms 16 & 17 were fabricated using 80 wt% PEGDA with a MW ratio (6:35 kDa) of 1:1 to improve the robustness of the resulting gel and minimize damage from the needle insertion required to create a fluid input port. Using the same glycerol solution, the phantom was perfused with a flow rate of 200 μL/min (to account for the larger channel diameters) and then coupled to the transducer with US gel and imaged using the same 3D color Doppler imaging and reconstruction protocol implemented for Phantom 16. These data were then used to calculate the magnitude of the fluid velocity, which was displayed as a 3D rendering fused with the B-mode data using the ParaView data analysis and visualization platform (Kitware, Inc., Clifton Park, NY) [43].

## Results and discussion

### Assessment of US properties and stability

US backscatter was generated at the boundary of each printed layer (Fig 3C), resulting from local density mismatches induced at hydrogel-layer boundaries (Fig 3B). In the optical phantoms with FITC-dextran, fluorescence intensity within each layer was inversely correlated with distance from the light source (Fig 3A), indicating that light exposure (i.e., from the projector source) was highest in the most proximal portion of each layer. Because extrusion printing is used, regions with the highest light exposure from one layer are inherently adjacent to regions with the lowest light exposure from the next layer, leading to discontinuities in light exposure through depth (Fig 3A). In B-mode images of matched phantoms, these exposure discontinuities were aligned with the US scattering generated at each layer boundary (Fig 3C), demonstrating that the "stacking" of photocured layers that occurs with extrusion printing produces local density discontinuities that result in hyperechoic regions.

Hyperechoic layer boundaries were clearly visible in Phantom 1 (printed with 200-μm layers) when the 50-MHz transducer was used; however, these distinct boundaries were blurred

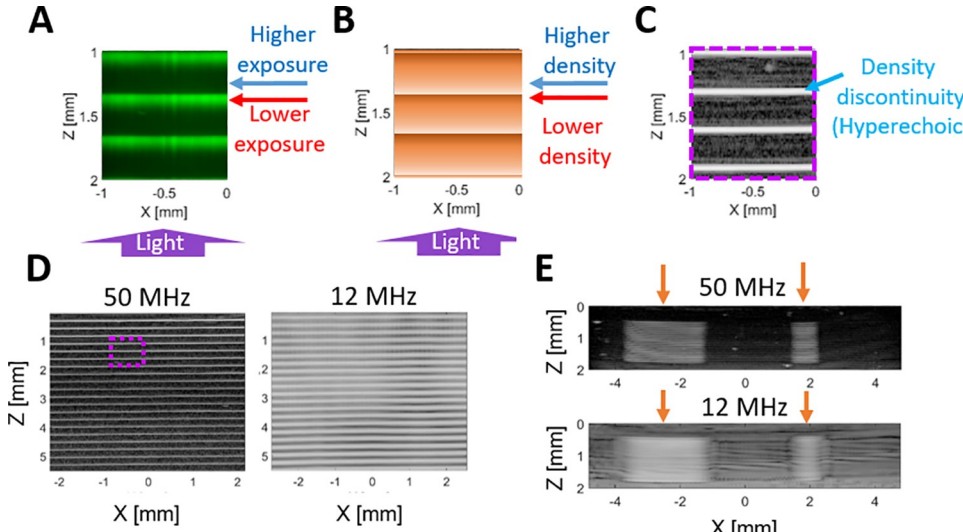

**Fig 3. Investigation of acoustic backscatter generation using optical and US imaging.** (**A**) Fluorescence imaging slice of Phantom 3 printed with 200-μm layers showing photobleached areas of high light exposure (blue arrow) and regions of low light exposure (red arrow). (**B**) Schematic of assumed phantom density, where dark regions indicate higher local density (blue arrow) and light regions indicate lower density (red arrow). The light source is below the phantom in the Z-direction (purple arrow). (**C**) Zoomed-in subsection (purple box in [**D**]) from Phantom 1 with 200-μm layers and imaged using 50-MHz B-mode US. Density mismatches appear as hyperechoic, while continuous-density regions appear anechoic. (**D**) 50-MHz (left) and 12-MHz (right) B-mode US images of Phantom 1 with 200-μm layers. (**E**) B-mode US images of Phantom 5 with two rectangular inclusions (orange arrows) printed with 50-μm layers at 50 MHz (top) and 12 MHz (bottom).

at 30 MHz and difficult to distinguish at all at 12 MHz due to the spatial averaging resulting from lower-frequency imaging (Fig 3D). For phantoms printed with 50-μm layers, the print-layer boundaries were spatially averaged even using a 50-MHz center frequency, and they were indistinguishable at 12 MHz (S2 Fig), indicating that such print-voxel size is well suited for fabricating phantoms intended for more typical clinical US frequencies (i.e., <12 MHz).

Different cure times in laterally adjacent voxels in the X-Y plane of the phantom produced different echogenicity levels (i.e., rectangular inclusions in Fig 3E). Quantifying the backscatter signal in regions of increased exposure in Phantoms 5–7 revealed a nonlinear effect of cure time on US signal, where the first few additional seconds (i.e., the leftmost printed targets in Fig 4A and 4B) generated a large increase in the US signal relative to the baseline. However, this backscatter signal increased and then leveled off after an additional exposure of 5 seconds (i.e., the rightmost printed inclusions; Fig 4C). This increase in US signal within the targets was noted at all three investigated imaging frequencies, and in phantoms with and without xanthan gum or silica (Phantoms 5–7). Phantoms with silica concentrations of 1 mg/mL (i.e., Phantom 8) provide scattering more consistent with speckle than phantoms with lower silica concentrations (i.e., 0.1 mg/mL; Phantom 7) or phantoms with no silica added (i.e., Phantom 6; S5 Fig). As part of future work, silica and xanthan gum concentrations can be optimized in combination with print-layer thickness (i.e., to ensure complete photocuring in the presence of additional scatterers) to ensure that silica particles do not settle and/or aggregate during the phantom photocuring process.

Because PEGDA hydrogels can degrade over time via hydrolysis [44], we tested the temporal stability of the backscatter signal they generate. The US signal from all 1.5-mm printed targets in Phantom 5 (Fig 4A) decreased over time (Fig 4C). However, the CNR remained relatively stable over 6 weeks of storage and imaging (Fig 4D). Focal regions of

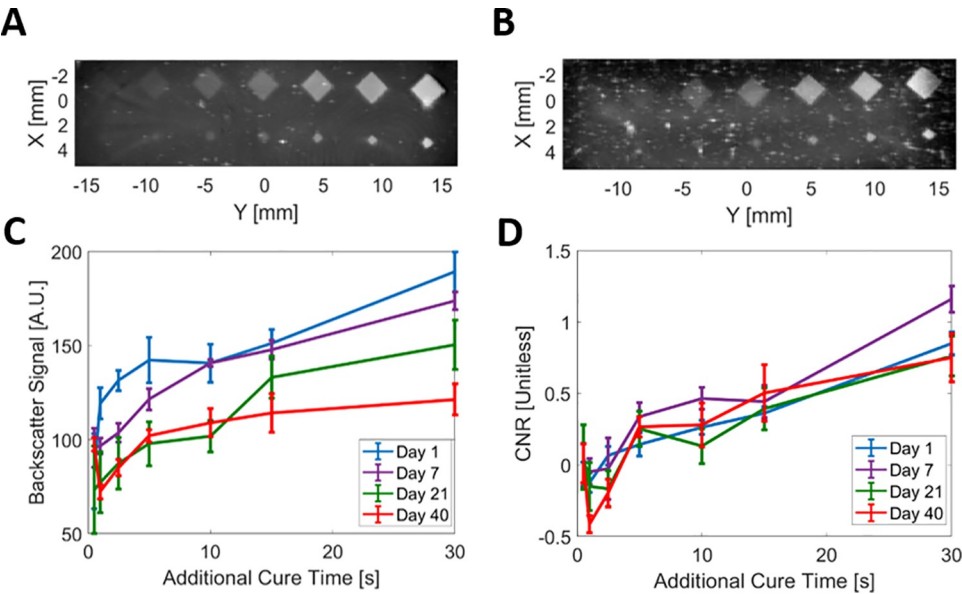

**Fig 4. Backscatter patterning and photocuring dependence.** C-scan US images of patterned phantoms (**A**) without silica particles (Phantom 4) and (**B**) with silica particles (Phantom 7; 0.1 mg/mL). (**C**) US backscatter signal and (**D**) CNR for each additional cure time over 6 weeks imaged at 30 MHz in Phantom 7. The data points for additional cure time in (**C**) and (**D**) each correspond to a unique printed target in the phantoms shown in (**A**) and (**B**).

hyperechogenicity (e.g., X,Y = 3,2 in Fig 4A) are likely a result of dust contamination during printing, an effect that can be mitigated by physically shielding the open print stage from the environment and/or by placing the stage in a (slight) negative-pressure vacuum hood. Morphologic changes of phantoms were also assessed immediately following printing and during storage. Hydrogels containing GelMA (Phantom 11) stored in PBS were morphologically most stable (S1 Table), experiencing an average (i.e., for both dimensions of the phantom body and lumen) absolute dimensional change compared to their CAD design of 5.2% (day of print) and 4.8% (31 days after print) at 4°C and 4.4% (day of print; this value was also used as the estimate for accuracy) and 6.6% (31 days after) at 25°C. Hydrogels without GelMA (Phantom 10) stored in PBS were less stable, experiencing an average absolute dimensional change compared to their CAD design of 12.4% (day of print) and 26.1% (31 days after print) at 4°C and 8.2% (day of print) and 21.1% (31 days after print) at 25°C. Hydrogels stored in DI water, regardless of storage temperature, immediately experienced significant (>20%) dimensional increases (i.e., both lumen and body) when compared to their initial CAD dimensions, an expected result given the osmotic imbalance between the PBS-based phantoms and (salt-free) DI water. Across all combinations, reproducibility measurements between matched phantoms differed by an average of 2.8%, while the average precision between measurements within the same phantom was 3.0%. Although print accuracy was estimated at 4.4%, such a metric can be difficult to characterize as interaction of the phantom with the measurement environment (e.g., PBS) can quickly cause changes from the original print geometry. In future work, we intend to investigate different PEG crosslinking chemistries that are more resistant to hydrolysis. For example, we previously demonstrated that PEG-diacrylamide is compatible with our fabrication technique [33], and this material has been shown to be resistant to hydrolysis [44].

We also characterized the group velocity through our hydrogel formulation (Phantom 9) to be 1527±1, 1523±2, & 1527±1 m/s for 2.25, 5, & 10MHz, respectively. This result is similar to work done by Aliabouzar *et al.*, which reported sound speed values between 1500–1600 m/s

for the same class of hydrogel [34]. This prior study also demonstrated that similar 3D-printed PEGDA-based samples as those investigated in our study present with US attenuation that is consistent with soft tissue, reporting average attenuation values of 0.54, 0.85, & 1.27 dB cm$^{-1}$ MHz$^{-1}$ when measured with center frequencies of 2.25, 5, 10 MHz, respectively.

## Elasticity imaging

Shear wave imaging data for the compliant elasticity phantom (Phantom 12) showed mean shear wave velocities between 1.6±0.1 and 2.3±0.1 m/s within regions of 5 and 12.5 seconds of cure time, respectively (S2 Table). These shear wave velocities correspond to Young's moduli of 7.6 and 15.6 kPa, respectively, assuming a linear, isotropic, elastic medium; these values are within the range reported for *in vivo* imaging of soft tissue [41]. The stiff elasticity phantom (Phantom 13) presented a similar trend of increasing shear wave velocity with cure time; however, Young's modulus estimates within the inclusions were significantly higher than typical soft tissue values, ranging from 72.9 to 123.6 kPa (S2 Table). In future studies, we will investigate the use of shorter photocuring times to establish a more precise relationship between cure time and stiffness; we will also examine the correlation between changes in US backscatter and stiffness as a function of photocuring time.

For the anisotropy elasticity phantom (Phantom 14), measured shear wave velocity (S2 Table) increased from 2.3±0.03 m/s to 2.6±0.08 m/s when the transducer was rotated from perpendicular (90˚) to parallel (0˚) with the printed column inclusions. These results yielded a significant 12% increase in shear wave velocity based on orientation, demonstrating transverse shear wave anisotropy within the phantom. Visualizations of the shear wave propagation over time for the 0˚ orientation can be seen in S3 Fig. Further development of this phantom model could prove useful for modeling tissues with striated fibers, such as muscle, as this phantom contained periodic anisotropic, chord-like structures which could mimic naturally occurring fiber orientation.

## 3D flow imaging

The goal in creating the small-channel flow phantom (Phantom 15) was to fabricate a phantom with channels approaching the order of size common for capillaries. The channel diameters were inconsistent throughout the extent due to differences in light scattering during photocuring, but diameters were measured as small as 60 μm (Fig 5A). Despite this inconsistency, all channels supported flow that tended to be laminar in nature upon investigation with quantitative analysis based on Doppler imaging.

In the serpentine flow phantom (Phantom 16), the printed channels sustained parabolic flow profiles (Fig 5C, 5D). However, flow profile inconsistencies occurred in some locations (i.e., yellow arrow in Fig 5B), corresponding to regions of physical phantom irregularities, as confirmed with B-mode US imaging. Representative 2D images of Doppler overlaid B-mode data from Phantom 16 are provided in S4 Fig. As the phantom generally sustained a symmetric parabolic flow pattern, we are confident in our ability to fabricate a phantom with predictable flow patterns, and in future work, we will investigate more complex flow geometries.

The tumor flow phantom (Phantom 17) demonstrates variable levels of US contrast (i.e., hypoechoic tumor relative to background) and branching channels that can support fluid flow. In Fig 6C, a 3D rendering of the magnitude of Doppler-estimated flow velocity is fused with an US B-mode rendering, showing the printed "tumor" in the central region. Cross-sectional flow profiles (rightmost images in Fig 6C) present expected fluid flow patterns, with the central region of the channel lumen experiencing the highest velocity flow and continuous flow being observed throughout the channel.

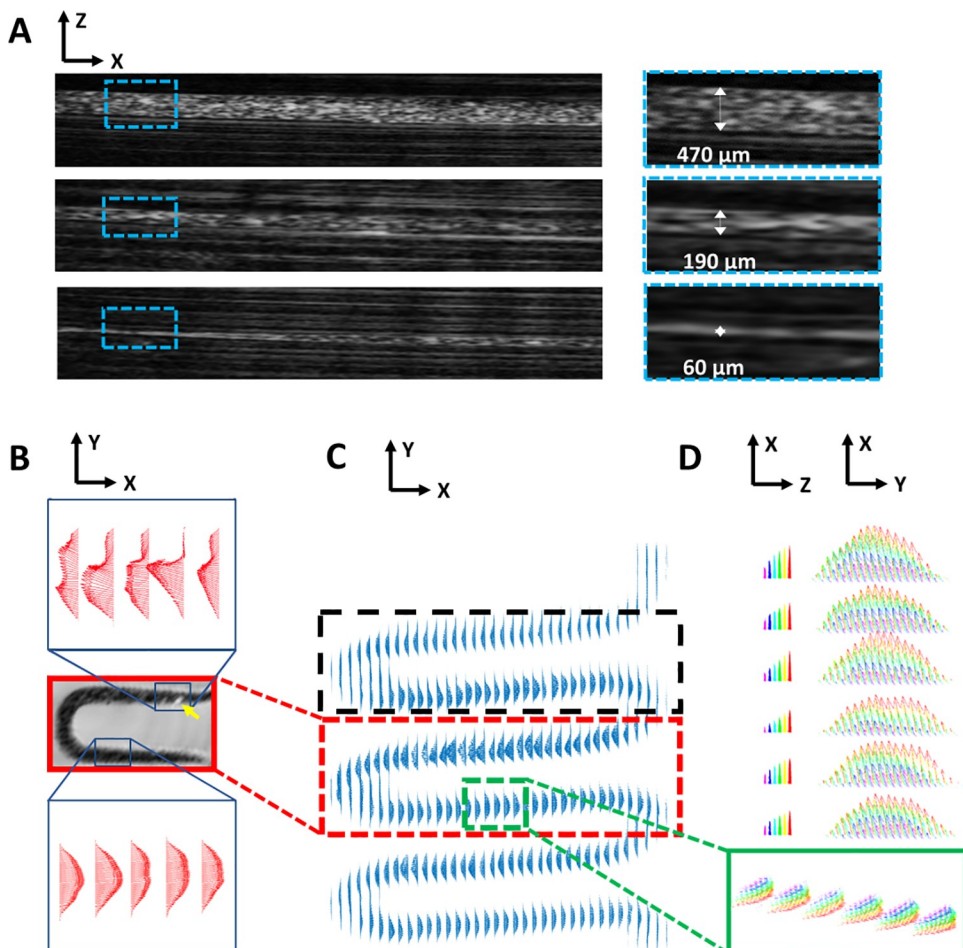

**Fig 5. Generation of phantoms with flow-supporting channels.** (**A**) B-mode US images (left) of Phantom 15 showing fluorescent microbeads injected into channels and zoomed-in regions (right; cyan dashed boxes) show channels as small as 60 μm remain open and support flow. (**B**) Zoomed-in views of two regions within the red box in (**C**). A physical imperfection (i.e., the hyperechoic point denoted by yellow arrow) within the fabricated channel caused disturbances in the symmetry of the parabolic flow profiles within this region (top); downstream from the imperfection, the flow returned to normal (bottom). Dashed black box indicates the location of the image cut-out shown in S4 Fig. (**C**) Doppler-derived flow velocity vectors through the serpentine channel (Phantom 16). (**D**) Zoomed-in view of the region in the green box in (**C**) showing a symmetric, parabolic velocity profile.

The fabrication method proposed in this work provides a framework for voxel-specific US phantoms, which can be designed to bear the physical and acoustic properties shared by many soft tissues. The process not only generates US phantoms with properties that can be modified via both light exposure and external additives, but it allows for the production of an entire phantom in a single print session regardless of complexity. Note that the maximal print size of phantoms can be increased by increasing the size of the print platform. Additionally, a gelatin block can be cast around finer-detail, printed hydrogels with conventional backfill techniques to increase effective imaging depth and ensure adequate transducer coupling to the larger (hybrid) phantom. In the future, the proposed 3D-printing platform could be utilized with patient-derived anatomical data from CT or MRI to integrate actual patient data when fabricating a well-characterized phantom. The platform's demonstrated viability for embedded living cells [33] also offers unique opportunities for highly precise and potentially more realistic

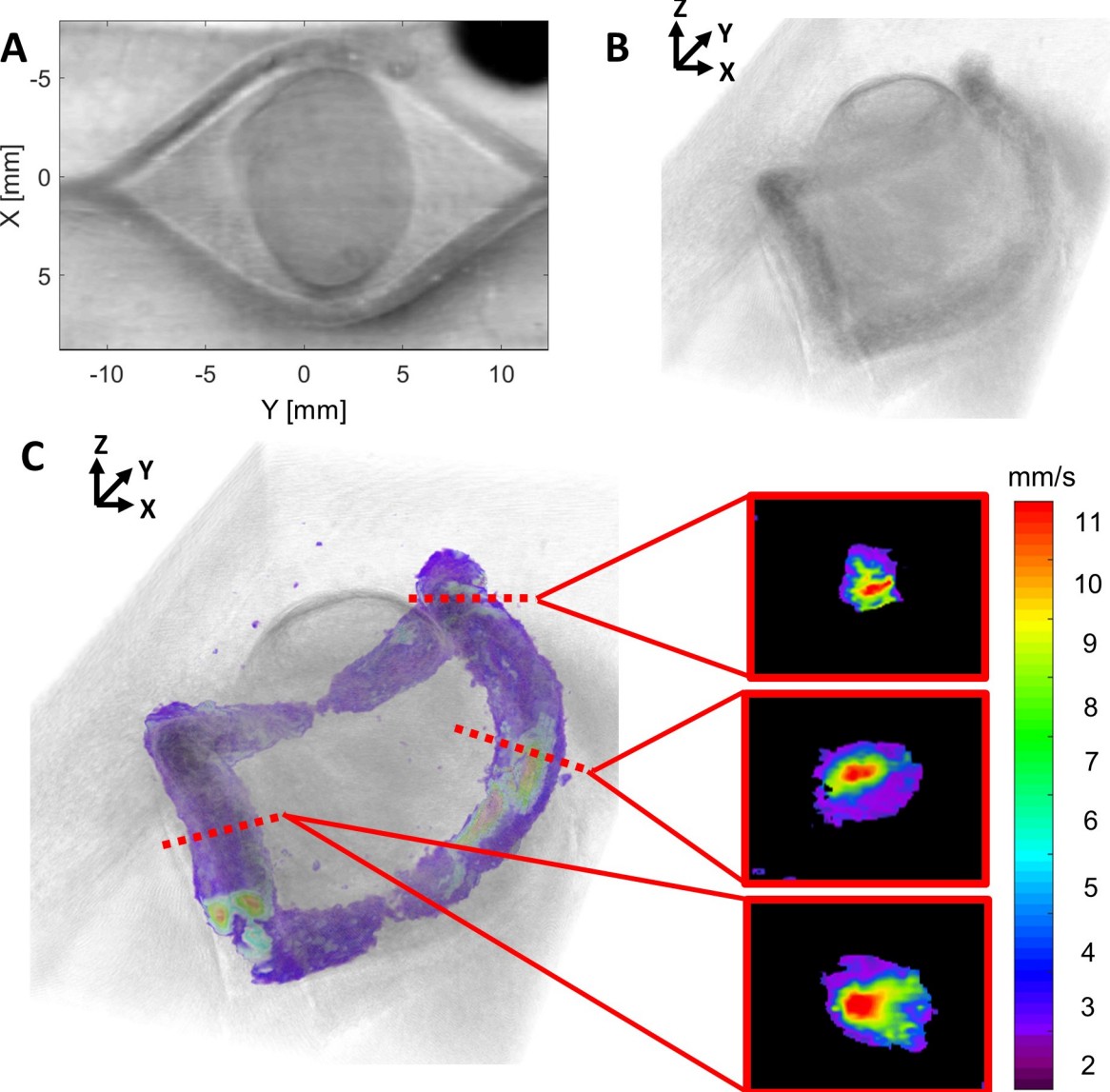

**Fig 6. 3D reconstruction of flow around a hypoechoic tumor region.** (**A**) B-mode C-scan of Phantom 17 showing a hypoechoic tumor inclusion flanked by two vessels. (**B**) 3D B-mode US rendering of Phantom 17 and (**C**) this rendering fused with the magnitude of Doppler-estimated flow velocity through the channel network; to the right, cross-sectional flow profiles are provided for three representative locations, indicated by the dashed red lines.

*in vitro* imaging studies. Ultimately, this US phantom fabrication process provides a first step toward making phantoms that are realistic enough to assist in the development and validation of the next generation of US-based functional and quantitative imaging methods.

## Conclusions

Our pSLA 3D printing technique is a viable approach for effectively fabricating phantoms for a broad range of US-mediated imaging applications. By modifying instructions for our 3D printing system, we can control the amount of light delivered to different regions within the hydrogel to manipulate both the local backscatter coefficient and stiffness while maintaining US

properties consistent with soft tissue. Ultimately, we demonstrated the capacity to fabricate US phantoms in a single, semi-automated process containing patterns of complex and customizable backscatter, regions of varied and anisotropic elasticity consistent with soft tissue, and open-channel networks that can mimic naturally occurring vasculature and support flow. Collectively, these results show that projection-based stereolithography shows tremendous promise in providing a next-generation US fabrication phantom platform.

## Supporting information

**S1 Fig. Function of custom Python script to incorporate differential curing within each layer.** (**A**) A Python script adds regions of secondary exposure to the primary background exposure to produce the illumination pattern for the printed object at each slice. (**B**) This process results in a final monolithic gel with different levels of photocuring.
(TIF)

**S2 Fig. Hyperechoic signal at print-layer boundaries.** B-mode images at (**A**) 12 MHz, (**B**) 30 MHz, & (**C**) 50 MHz of Phantom 1 with 200-μm layers showing the hyperechoic signal presenting at layer interfaces. B-mode images at (**D**) 12 MHz, (**E**) 30 MHz, & (**F**) 50 MHz of 50-μm layer Phantom 2.
(TIF)

**S3 Fig. Representative 2D shear wave images.** Images of US-based axial displacement estimates showing shear wave propagation (wave fronts identified with white arrows) at four time-points following an acoustic radiation force impulse in an anisotropic elasticity phantom (0˚ in Phantom 14).
(TIF)

**S4 Fig. Representative 2D Doppler images.** (**A**) Doppler data overlaid on B-mode images and (**B**) cut-out of the flow-velocity vector data from Fig 5C denoting the two imaging planes (distinguished by orange or purple dashed lines/arrows) shown through channels in Phantom 16 with opposite flow directions.
(TIF)

**S5 Fig. Representative 2D B-mode images with varying silica concentration.** B-mode images at 30 MHz of phantoms with 50-μm layer thickness and (**A**) only xanthan gum (0.833 mg/mL; Phantom 6), (**B**) xanthan gum (0.833 mg/mL) and 0.1 mg/mL silica particles (Phantom 7), and (**C**) xanthan gum (0.833 mg/mL) and 1 mg/mL silica particles (Phantom 8).
(TIF)

**S1 Table. Percent differences over time of hydrogels stored in PBS at 4˚C and 25˚C when compared to designed CAD dimensions.**
(DOCX)

**S2 Table. Results from US elasticity imaging.**
(DOCX)

## Acknowledgments

We thank the open-source projects that facilitated this work, including the NIH ImageJ, Blender.org, ParaView, and the NIH 3D print exchanges. We would also like to thank the MD Anderson Office of Scientific Publications for their help in preparing the manuscript and Mark Palmeri for providing shear wave imaging and analysis code.

## Author Contributions

**Conceptualization:** Mark D. Pagel, Jordan S. Miller, Richard R. Bouchard.

**Formal analysis:** Samantha J. Paulsen, Trevor M. Mitcham, James Long, Collin J. Harlan.

**Funding acquisition:** Jordan S. Miller, Richard R. Bouchard.

**Investigation:** Samantha J. Paulsen, Trevor M. Mitcham, Charlene S. Pan, James Long, Bagrat Grigoryan, Daniel W. Sazer, Collin J. Harlan, Kevin D. Janson.

**Methodology:** Samantha J. Paulsen, Trevor M. Mitcham, Bagrat Grigoryan, Daniel W. Sazer, Kevin D. Janson, Jordan S. Miller, Richard R. Bouchard.

**Project administration:** Jordan S. Miller, Richard R. Bouchard.

**Resources:** Samantha J. Paulsen, Kevin D. Janson.

**Software:** Trevor M. Mitcham.

**Supervision:** Jordan S. Miller, Richard R. Bouchard.

**Validation:** Samantha J. Paulsen.

**Visualization:** Samantha J. Paulsen, Trevor M. Mitcham.

**Writing – original draft:** Samantha J. Paulsen, Trevor M. Mitcham.

**Writing – review & editing:** Samantha J. Paulsen, Trevor M. Mitcham, James Long, Collin J. Harlan, Mark D. Pagel, Jordan S. Miller, Richard R. Bouchard.

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
