## [Decision Letter · Decision Letter 0]

21 Sep 2020

PONE-D-20-14537

Projection-based stereolithography for direct 3D printing of heterogeneous ultrasound phantoms

PLOS ONE

Dear Dr. Bouchard,

Thank you for submitting your manuscript to PLOS ONE. After careful consideration, we feel that it has merit but does not fully meet PLOS ONE’s publication criteria as it currently stands. Therefore, we invite you to submit a revised version of the manuscript that addresses the points raised during the review process.

Both reviewers find the manuscript potentially valuable for ultrasound imaging research and development, especially for anisotropic tissues. However, both reviewers also pointed out missing performance analysis (accuracy and precision) of the proposed printing method given the readily available ground truth from the design models. 

Please present how accurate, precise, reproducible and robust the proposed stereolithography method for printing complex tissue-mimicking phantoms for ultrasound studies in a quantitative way. In the present manuscript, no quantitative analysis is provided. In short, adding an in-depth performance analysis of the proposed method will significantly strengthen the paper.As presented in the manuscript, the proposed method should be versatile as evidenced by 15 different types of phantoms manufactured, but the way they were presented made the manuscript less focused as Reviewer 1 suggested. Therefore, authors may consider to reorganize the manuscript. For instance, categorize phantoms according to tissue properties (backscattering (speed of sound, angle dependence, density), elasticity (purely elastic or even viscoelastic), and geometry (thin layer, bulk, hollow cylindrical or vasculature)) that can be assessed by current ultrasound imaging techniques. Then, elaborate the importance of these tissue properties in respective ultrasound clinical applications. Talk about challenges of meeting required tissue properties using existing ultrasound phantom-making methods in the literature. Lastly, summarize how your proposed method tackled those challenges and can guide potential readers realistic tissue-mimicking phantom design and fabrication.Please address Reviewer 1's detailed comments as far as possible. Meanwhile, make study limitations clearer.Please submit your revised manuscript by Nov 05 2020 11:59PM. If you will need more time than this to complete your revisions, please reply to this message or contact the journal office at plosone@plos.org. Please include the following items when submitting your revised manuscript:
A rebuttal letter that responds to each point raised by the academic editor and reviewer(s). You should upload this letter as a separate file labeled 'Response to Reviewers'.A marked-up copy of your manuscript that highlights changes made to the original version. You should upload this as a separate file labeled 'Revised Manuscript with Track Changes'.An unmarked version of your revised paper without tracked changes. You should upload this as a separate file labeled 'Manuscript'.

We look forward to receiving your revised manuscript.

Kind regards,

Wei-Ning Lee

Academic Editor

PLOS ONE

Journal Requirements:

3. Please note that in order to use the direct billing option the corresponding author must be affiliated with the chosen institute. Please respond by return e-mail so that we can amend your submission or remove this option. We can make any changes on your behalf.

4. Please include a copy of Table 2 which you refer to in your text on page 9 and 10.

Reviewers' comments:

Reviewer's Responses to Questions

**Comments to the Author**

1. Is the manuscript technically sound, and do the data support the conclusions?

Reviewer #1: No

Reviewer #2: Yes

2. Has the statistical analysis been performed appropriately and rigorously? 

Reviewer #1: No

Reviewer #2: Yes

3. Have the authors made all data underlying the findings in their manuscript fully available?

Reviewer #1: Yes

Reviewer #2: Yes

4. Is the manuscript presented in an intelligible fashion and written in standard English?

Reviewer #1: Yes

Reviewer #2: Yes

5. Review Comments to the Author

Reviewer #1: The goal of the reported phantom work is relevant to quantitative ultrasound methods. Its main innovation is to demonstrate a single-session 3D printing approach for a variety of ultrasound applications that may simplify phantom development and allows control of backscatter and stiffness at the voxel level.

The presentation is not focused, however, rather several different phantom designs for different potential US applications are demonstrated but the rationale for these designs is not offered. Most appear to be intended for high frequency US imaging (still a small part of ultrasonography) but this is also not stated. It is argued that the proposed printing method has advantages over other phantom methods but these advantages are not clearly stated or demonstrated.

Perhaps it would help to focus the studies if measurements could be made to determine how accurately and precisely the intended phantom(s) specifications coded in the control software were rendered in the printed results. This seems to be the key question.

The manuscript is more in the form of a demonstration report or descriptive technical note rather than a succinct research paper. A number of fabrication factors that affect acoustic properties were examined but accuracy and precision of the measurements are not reported.

For production of a reference phantom or one mimicking tissue properties, there is a need to demonstrate the stability of phantom properties in terms of effects of room temperature, acoustic radiation force, humidity, time, etc. How reproducible are these properties for duplicate phantoms?

The authors argue that their 3D printed methods are more repeatable and reproducible that ordinary methods, one of the hallmarks of the QIBA effort. However, no data showing precision of the phantom properties is provided. What is the precision of their method for the acoustic properties that were measured? Was more than one phantom with the same technique printed and compared? What is the limit on spatial accuracy of the printing process?

The short title is the same as the full title, although good alternatives are possible, e.g., “Stereolithography 3D printing of ultrasound phantoms.”

The number of references may be excessive as many do not directly support the methods in the manuscript.

Line 19. The abstract summarizes the work and its purpose although the key question that was answered is not stated.

Line 48. The insufficiency of current phantoms depends on the QIB, the tissue of interest and the intended application. Elastography and flow are not the only QIBs of interest. For example, quantitative backscatter, attenuation, structure function, form factor, etc., using reference phantom methods are quite robust in liver, muscle and other (admittedly) more homogeneous organs. These methods depend on cellular-level structures, which would require adding scattering material to the 3D printing as described here.

Line 53. It may help to clearly state what is the purpose of having complex phantoms that approximate anatomy, function and mechanical properties of real tissues. How would they be used for QIB development? How close is good enough? The data presented do not appear to actually address anthropomorphic organ-mimicking phantoms anyway. Clearly state what they add that cannot be accomplished with simpler QIB phantoms that permit isolation of tissue properties independently or in simpler combinations.

Line 58. The quote from Ref 10 is referring to precision of QIB measurements specifically in phantoms, which may be overestimated. It is not referring to precision in patients for which many confounders are present but not present in phantoms. How do more complex phantoms contribute to solving this problem?

Line 112. “single fabrication step” should probably be changed to “single fabrication session.”

Line 128. “degree necessary” requires more explanation. Perhaps the current limits on printed voxel size and US field of view should be summarized somewhere.

Line 149. What are the x and y dimensions of the stage? Is this the limiting factor for the size of the phantom? All of the phantoms in Fig 2 are very small compared to the fields of view of clinical transducers. In addition, the B-mode images of the phantoms were made at nominally 12-50 MHz, so perhaps it should be stated early in the manuscript that the intended use for the phantoms is for high frequency applications. Can a phantom be fabricated that would contain the entire 3D dimensions of the human transducer? If so, what printing time might be required for Phantom A, if it were made a more useful size of 10 cm L x 10 cm W x 10 cm H? The authors state a printing rate of 3 cm depth per hour (at what x and y dimension?). What is the duration of a complete session needed to fully print a phantom that would encompass the full US transducer 3D field of view?

Line 157. Is this discontinuity interface a discrete boundary or is there also a gradient? To the same point, is it possible to create density gradients or only discrete layers?

Line 180. Rehydration and “swelling” of the phantom material appear to introduce a potential repeatability problem. Do you have data to support it does not?

Table 1. What was the total printing time to produce each phantom? It would help the reader to understand the advantages you claim for the single-session 3D printing versus other methods.

Line 197. You reference phantoms 8, 15 and 16. I do not see phantom 16 in this manuscript. What are the different concentrations of silica? How did backscatter coefficient vary with concentration and at what frequencies? How does the presence of the silica affect printing process and needed light exposure time compared to no silica in the formula?

Line 228 and 406. Since these layers are essentially specular reflectors, it may be confusing to refer to backscatter in this context especially since the phantoms have silica scatterers added to the formula. Perhaps you can refer to the effect of exposure time as changing “reflectivity” of acoustic impedance of the layers. The methods for measurement of “backscatter” are not described and the few results are in arbitrary units. Were these measures of pixel “echogenicity” from the B-modes? Ordinarily this incorporates many poorly controlled variables including gain, transmit power, TGC, grey-level LUT, position in FOV, etc. Sound speed estimates were made with the A-line RF signal; was “backscatter” as well?

Line 333. At 30 MHz, the transit time measurement should be robust while the caliper thickness error for 10 mm might contribute significant uncertainty. What is the accuracy and precision of both measurements in a homogeneous material with known speed of sound and thickness (e.g., nylon, acrylic, etc.)? How many independent measures of thickness and time were made?

Lines 294 and 342. Although you reference your methods as previously published, it would still be valuable to include the important experimental details here, such as frequency.

Line 390. Mean +/- SD indicates that you made multiple independent measures of each property. How many for each?

Line 405. Since the sequential printing procedure for each layer produces acoustic interfaces (mis-matches), does this present a problem for phantom design making it difficult to make large homogeneous regions?

Line 498. The shear wave speed results show promise for constructing anisotropic phantoms, at least in two dimensions. Researchers in muscle, tendon and nerve with quantitative US will find this has interesting potential for this work.

Line 560. The conclusions are appropriate for the material presented although they are very general.

Fig 2C. The spherical inclusions are stated to be 5 mm in diameter. What does “0-5 mm” mean here?

Fig 4 A and B. What is the origin of the dramatic non-uniformities?

Ref 44 is incomplete and it is not clear what type of publication this is (patent, book, etc.).

Ref 52 and 53 appear to be (almost) duplicates, referring to the same publication.

Reviewer #2: The authors present interesting developments in the realization and characterization of 3D printed US phantoms. Several samples and designs were fabricated and analyzed, while varying manufacturing parameters. Overall a very nice work that should be of interest to our readers.

One question, that perhaps this reviewer missed, is did the authors measure the as-printed feature dimensions in comparison to their original CAD model? Some understanding of the geometric fidelity of this process, if available, might also add value for the generation of complex 3D phantom structures.

6. PLOS authors have the option to publish the peer review history of their article (what does this mean?). If published, this will include your full peer review and any attached files.

Reviewer #1: No

Reviewer #2: No

---

## [Author Response · Author response to Decision Letter 0]

14 Jul 2021

We would like to thank the editor as well as both reviewers for their detailed comments on our work. We have addressed all of the points included, and we believe that it has strengthened our manuscript. A few brief responses are in order to address the comments from the editor (editor's comment in quotations with our response immediately following):

1. "Please present how accurate, precise, reproducible and robust the proposed stereolithography method for printing complex tissue-mimicking phantoms for ultrasound studies in a quantitative way. In the present manuscript, no quantitative analysis is provided. In short, adding an in-depth performance analysis of the proposed method will significantly strengthen the paper."

We have now presented data regarding the accuracy, precision, stability, and reproducibility of our phantoms, as detailed in the reviewer reply.

2. "As presented in the manuscript, the proposed method should be versatile as evidenced by 15 different types of phantoms manufactured, but the way they were presented made the manuscript less focused as Reviewer 1 suggested. Therefore, authors may consider to reorganize the manuscript. For instance, categorize phantoms according to tissue properties (backscattering (speed of sound, angle dependence, density), elasticity (purely elastic or even viscoelastic), and geometry (thin layer, bulk, hollow cylindrical or vasculature)) that can be assessed by current ultrasound imaging techniques. Then, elaborate the importance of these tissue properties in respective ultrasound clinical applications. Talk about challenges of meeting required tissue properties using existing ultrasound phantom-making methods in the literature. Lastly, summarize how your proposed method tackled those challenges and can guide potential readers realistic tissue-mimicking phantom design and fabrication."

We would like to thank the editor for her suggestions; we have reorganized our manuscript in an order to improve clarity and focus. We have also expanded on the importance of the tissue properties we selected, the areas where current phantoms struggle to match our method, and how our method could aid other researchers in fabricating tissue-mimicking phantoms.

3. "Please address Reviewer 1's detailed comments as far as possible. Meanwhile, make study limitations clearer."

We have addressed all of reviewer 1's comments, and in doing so strengthened our work. Study limitations have also been addressed in the work. 

Detailed replies to the specific comments from both reviewers can be found within the reviewer response document.

---

## [Decision Letter · Decision Letter 1]

14 Sep 2021

PONE-D-20-14537R1Projection-based stereolithography for direct 3D printing of heterogeneous ultrasound phantomsPLOS ONE

Dear Dr. Bouchard,

Thank you for submitting your manuscript to PLOS ONE. After careful consideration, we feel that it has merit but does not fully meet PLOS ONE’s publication criteria as it currently stands. Therefore, we invite you to submit a revised version of the manuscript that addresses the points raised during the review process.

 The authors have improved the manuscript significantly. As the second round reviewer pointed out, the study targeted at ultrasound imaging applications. The authors are advised to provide acoustic properties of the fabricated phantoms and discuss about fabrication reproducibility of the desired acoustic properties. 

We look forward to receiving your revised manuscript.

Kind regards,

Wei-Ning Lee

Academic Editor

PLOS ONE

Journal Requirements:

Reviewers' comments:

Reviewer's Responses to Questions

**Comments to the Author**

1. If the authors have adequately addressed your comments raised in a previous round of review and you feel that this manuscript is now acceptable for publication, you may indicate that here to bypass the “Comments to the Author” section, enter your conflict of interest statement in the “Confidential to Editor” section, and submit your "Accept" recommendation.

Reviewer #3: (No Response)

2. Is the manuscript technically sound, and do the data support the conclusions?

Reviewer #3: Partly

3. Has the statistical analysis been performed appropriately and rigorously? 

Reviewer #3: N/A

4. Have the authors made all data underlying the findings in their manuscript fully available?

Reviewer #3: Yes

5. Is the manuscript presented in an intelligible fashion and written in standard English?

Reviewer #3: Yes

6. Review Comments to the Author

Reviewer #3: This manuscript describes a 3D-printing technique of heterogeneous ultrasound phantoms based on stereolithography. The stiffness of the phantom can be controlled from 7 to >120 kPa. Blood-mimicking fluid flow can also be supported by this kind of phantom. This work is well written, and this kind of phantom could be useful if it can be well encapsulated. However, the author should provide more details on the ultrasound properties of the phantom.

General comments

1.This kind of phantom was mainly used in ultrasound imaging study, therefore, the ultrasound properties of the phantom, such as speed of sound and attenuation, should be presented.

2.The 2D imaging of Bmode, color Doppler, shear wave elastography should be provided in this study.

3.Typically, scatterer is important for ultrasound imaging, however, uniform scatterer distribution can not be seen in the figure 3 and figure 4, it should be discussed. In addition, some strong reflection like bubbles can been seen in figure 4, the author should also discuss it and explain how to avoid the the bubbles in phantom cooking.

4.Is this phantom water based? Should the phantom be used and stored in water? Did the author consider any preservative in this study?

7. PLOS authors have the option to publish the peer review history of their article (what does this mean?). If published, this will include your full peer review and any attached files.

Reviewer #3: No

---

## [Author Response · Author response to Decision Letter 1]

13 Oct 2021

Editor comments:

The authors have improved the manuscript significantly. As the second round reviewer pointed out, the study targeted at ultrasound imaging applications. The authors are advised to provide acoustic properties of the fabricated phantoms and discuss about fabrication reproducibility of the desired acoustic properties. 

We would like to thank the editor for their consideration of our work. We have made changes to the manuscript to better demonstrate the wide range of acoustic properties possible within our phantoms. We have additionally addressed all individual reviewer comments in our formal reviewer reply which is attached within the submission.

---

## [Editor Report · Decision Letter 2]

17 Nov 2021

Projection-based stereolithography for direct 3D printing of heterogeneous ultrasound phantoms

PONE-D-20-14537R2

Dear Dr. Bouchard,

We’re pleased to inform you that your manuscript has been judged scientifically suitable for publication and will be formally accepted for publication once it meets all outstanding technical requirements.

Kind regards,

Wei-Ning Lee

Academic Editor

PLOS ONE
---

## [Editor Report · Acceptance letter]

1 Dec 2021

PONE-D-20-14537R2 

Projection-based stereolithography for direct 3D printing of heterogeneous ultrasound phantoms 

Dear Dr. Bouchard:

I'm pleased to inform you that your manuscript has been deemed suitable for publication in PLOS ONE. Congratulations! Your manuscript is now with our production department. 

Kind regards, 

on behalf of

Dr. Wei-Ning Lee 

Academic Editor

PLOS ONE